# Prostate Cancer Treatment-Related Toxicity: Comparison between 3D-Conformal Radiation Therapy (3D-CRT) and Volumetric Modulated Arc Therapy (VMAT) Techniques

**DOI:** 10.3390/jcm11236913

**Published:** 2022-11-23

**Authors:** Fabrizio Tonetto, Alessandro Magli, Eugenia Moretti, Andrea Emanuele Guerini, Annarita Tullio, Chiara Reverberi, Tino Ceschia, Luigi Spiazzi, Francesca Titone, Agnese Prisco, Marco Andrea Signor, Michela Buglione, Gioacchino De Giorgi, Marco Trovò, Luca Triggiani

**Affiliations:** 1Department of Radiation Oncology, University General Hospital, 33100 Udine, Italy; 2Department of Medical Physics, University General Hospital, 33100 Udine, Italy; 3Department of Radiation Oncology, ASST Spedali Civili and Brescia University, 25100 Brescia, Italy; 4Hygiene and Clinical Epidemiology Unit, University General Hospital, 33100 Udine, Italy; 5Department of Urology, University General Hospital, 33100 Udine, Italy

**Keywords:** prostate cancer, image-guided radiotherapy (IGRT), volumetric modulated arc therapy (VMAT), treatment-related toxicity

## Abstract

**Simple Summary:**

Despite the potential benefits of conformal radiotherapy techniques, such as volumetric modulated arc therapy (VMAT), for the treatment of prostate cancer, their actual clinical impact has been under-investigated. This paper illustrates the results of a mono-institutional registry trial, with the aim to compare gastrointestinal (GI) and genitourinary (GU) toxicity rates among patients that underwent radical radiotherapy for prostate cancer using 3D conformal radiation therapy (3D-CRT) or VMAT. Overall, we enrolled 83 consecutive patients, treated with a prescribed dose of 70 Gy in 28 fractions (50.6% with 3D-CRT and 49.4% with VMAT). After a median follow-up of 77.26 months, the patients in the VMAT group had a lower (although not statistically significant) rate of late GI and GU toxicity. Rates of G ≥ 2 toxicities were low among the whole cohort of these patients treated with IGRT. VMAT allowed for a dose reduction to the rectum and bladder for the large majority of the parameters; nonetheless, the only parameter correlated with late GI toxicity G ≥ 2 was a rectal dose limit V66 > 8.5%.

**Abstract:**

Objective: This paper illustrates the results of a mono-institutional registry trial, aimed to test whether gastrointestinal (GI) and genitourinary (GU) toxicity rates were lower in localized prostate cancer patients treated with image-guided volumetric modulated arc therapy (IG-VMAT) compared to those treated with IG-3D conformal radiation therapy (IG-3DCRT). Materials and Methods: Histologically proven prostate cancer patients with organ-confined disease, treated between October 2008 and September 2014 with moderately hypofractionated radiotherapy, were reviewed. Fiducial markers were placed in the prostate gland by transrectal ultrasound guide. The prescribed total dose was 70 Gy in 28 fractions. The mean and median dose volume constraints for bladder and rectum as well as total volume of treatment were analyzed as potentially prognostic factors influencing toxicity. The Kaplan–Meier method was applied to calculate survival. Results: Overall, 83 consecutive patients were included. Forty-two (50.6%) patients were treated with 3D-CRT and 41 (49.4%) with the VMAT technique. The median follow-up for toxicity was 77.26 months for the whole cohort. The VMAT allowed for a dose reduction to the rectum and bladder for the large majority of the considered parameters; nonetheless, the only parameter correlated with a clinical outcome was a rectal dose limit V66 > 8.5% for late GI toxicity G ≥ 2 (*p* = 0.045). Rates of G ≥ 2 toxicities were low among the whole cohort of these patients treated with IGRT. The analysis for rectum dose volume histograms (DVHs) showed that a severe (grade ≥ 2) late GI toxicity was related with the rectal dose limit V66 > 8.5% (*p* = 0.045). Conclusions: This study shows that moderate hypofractionation is feasible and safe in patients with intermediate and high-risk prostate cancer. Daily IGRT may decrease acute and late toxicity to organs at risk and improve clinical benefit and disease control rate, cutting down the risk of PTV geographical missing. The adoption of VMAT allows for promising results in terms of OAR sparing and a reduction in toxicity that, also given the small sample, did not reach statistical significance.

## 1. Introduction

The implementation of radiotherapy dose escalation for localized prostate cancer patients provided clinical benefits by reducing biochemical failure and tumor progression [1,2]. Nevertheless, it is often challenging to increase the total dose to the target volume without exceeding the surrounding tissues’ tolerance dose. Radiobiological studies demonstrated that prostate cancer cells and pelvic healthy tissues have different radiosensitivities: the prostate cancer α/β ratio is close to 1.5 Gy, while the normal pelvic organs α/β ratio is nearly 3 Gy [3,4]. Compared to the 3D-conformal radiation therapy (3D-CRT), the intensity modulated radiation therapy (IMRT) technique delivers higher conformal doses to the planning target volume (PTV) and reduces the total dose to the organs at risk (OAR), achieving a better toxicity rate [5,6,7]. In addition, the combination of the image-guided radiation therapy (IG-RT) technique and moderate hypofractionated (HF) radiotherapy regimens further reduces the risk of radio-induced toxicity and improves the therapeutic index ratio [5,8,9,10,11,12,13,14,15,16,17,18,19,20,21,22,23,24,25].

This paper illustrates the results of a mono-institutional registry trial, aimed to test whether gastrointestinal (GI) and genitourinary (GU) toxicity rates were lower in patients treated with IG-VMAT compared to those treated with IG-3DCRT. Moreover, we investigated possible prognostic factors correlated with treatment-related side effects, particularly with late toxicity.

## 2. Materials and Methods

### 2.1. Patients Selection

Between October 2008 and September 2014, histologically proven prostate cancer patients with organ-confined disease (cT1-2, cN0, cM0) were included in this prospective phase II clinical trial. This research has been approved by the University Hospital Santa Maria della Misericordia of Udine, Italy, ethic committee on 30 September 2008 (Institutional Review Board number 238/30.09.2008), and written informed consent was obtained from all patients. Exclusion criteria were: ECOG performance status > 2 and prior radiotherapy treatment on pelvic sites. At the patients’ first clinical access, a physician collected data on patient medical history, clinical evaluation, PSA level, Gleason score (GS), clinical tumor staging, and prognostic classification according to National Comprehensive Cancer Network (NCCN) guidelines. Androgen deprivation therapy (ADT) was prescribed if indicated. All patients were screened before starting radiotherapy with the International Prostatic Symptoms Score (IPSS) questionnaire for urinary symptoms and quality of life: hypofractionated radiotherapy was proposed only for patients with an IPSS score of 15 or less. Radiotherapy treatment was delivered with 3DCRT technique until September 2012 and, from then on, patients were treated with VMAT technique. Patients were assigned to 3DCRT or VMAT radiotherapy treatment by temporal criterion.

### 2.2. IGRT Strategy

In order to reduce prostate gland displacement, a standard protocol for rectum and bladder filling was applied. Patients were instructed to use a fleet enema the evening before and the morning of the treatment simulation session. All patients underwent CT scanning with moderately full bladders and empty rectums. If the bladder (including filling) exceeded 100 mL, we proceeded with re-scanning. Furthermore, patients were instructed to have a comfortably full bladder and an empty rectum before each RT delivery session. Two months before treatment, 3 gold non-magnetic and hypoallergenic Fiducial Markers-CIVCO (FM), were placed in the prostate gland by transrectal ultrasound. IGRT was based on FM recognition and realignment by electronic portal imaging with 3DCRT and CBCT in combination with VMAT.

### 2.3. Radiotherapy Planning, Treatment Characteristics, and Follow Up

CT-scan simulation was conducted acquiring 3 mm slices imaging. All patients were positioned supine with arms crossed over the chest and legs immobilized by using the Combifix system. Treatment was delivered 5 days/weekly, the prescribed total dose to the PTV was 70 Gy in 28 fractions (2.5 Gy/each fraction). In the 3D-CRT group, the dose was prescribed so that 100% of PTV was covered by 95% of the dose/(V_95_ = 100%). In the 3D-CRT protocol, two different clinical target volumes (CTVs) were contoured: the CTV_1_ included both the prostate gland and seminal vesicles, and the CTV_2_ covered the prostate gland only. The PTV was created by adding a 5 mm isotropic expansion to the CTV. Treatment was delivered in 2 phases in the 3D-CRT group: PTV_1_ received a total dose of 50 Gy and sequentially a dose boost-up to 70 Gy was delivered to PTV_2_. A total dose of 70 Gy was prescribed to a single PTV obtained by a 5 mm isotropic expansion of a CTV encompassing both prostate and seminal vesicles in the VMAT group. The dose was prescribed so that at least 95% of PTV was covered by 100% of the dose (V100% ≥ 95%) in the VMAT group. The 3D-CRT was delivered with 6 MV photon beams, by Varian Clinac 600C linear accelerator equipped with Millenium 120 Multileaf Collimator (MLC), typically with 5 coplanar static beams with gantry positions at 270°, 315°, 0°, 45°, and 90°. Patients enrolled in the VMAT protocol were treated with a couple of 6 MV full arcs, delivered with iX Varian Linac, equipped with Millenium 120 MLC. Patients were followed-up every 3 months for the first year and then every 6 months for 5 years. The follow-up visit consisted in clinical evaluation, PSA value, and blood tests. The toxicity events were scored according to Common Terminology Criteria for Adverse Events (CTCAEs v 3.0).

### 2.4. Statistical Analysis

The aim of this study was to test whether the rate of acute (defined as events presenting in the first three months after radiotherapy) and late (defined as events presenting more than 3 months after radiotherapy ended) GI and GU toxicity in patients with localized prostate cancer was lower in the group of patients treated with VMAT technique compared to the 3D-CRT group. Mean and median dose volume constraints for bladder and rectum and total volume of treatment were analyzed as potentially prognostic factors influencing toxicity.

The study population features were investigated performing descriptive statistics on categorical and numerical variables. For categorical variables we considered frequency distributions, and for numerical variables we considered mean, median, interquartile range, standard deviation, 25° and 75° percentile, as well as minimum and maximum values. Kolmogorov–Smirnov test was performed to verify normal distribution of numerical variables.

For studying acute toxicity profiles, we used the Kaplan–Meier method, the Log-rank test for group comparisons, and the Cox regression model for regression analyses.

Late toxicity analysis was performed applying an informative censoring to deceased patients without experiencing late toxicity or relapse and running the cumulative incidence function (CIF) method for competing risks. For group comparisons, we used the homogeneity test of Gray (CIF, *p*-value). We conducted regression analyses performing the Cox regression model modified by Fine and Gray, obtaining the subdistribution hazards (SDH, *p*-value) univariate and adjusted radiotherapy method.

Cancer-specific survival (CCS) was defined as duration from the start of radiotherapy until death due to cancer; biochemical relapse-free survival (b-RFS) was characterized by Phoenix Criteria as a PSA level reaching 2 ng/mL more than the lowest patients’ PSA value post-treatment (PSA nadir).

For the definition of CSS and b-RFS, we applied an informative censoring to deceased patients for non-cancer causes (for CCS) or deceased without experiencing relapse (for b-RFS) and running the cumulative incidence function (CIF) method for competing risks. For group comparisons, we used the homogeneity test of Gray (CIF, *p*-value). We conducted regression analyses performing the Cox regression model modified by Fine and Gray, obtaining the subdistribution hazards (SDH, *p*-value) univariate and adjusted radiotherapy method.

The statistical analysis was performed by using SAS© software version 9.4 (SAS institute, Inc., Cary, NC, USA) and R 3.4.2. The significance level was set at 0.05.

## 3. Results

In the study period, 83 consecutive patients were included. A total of 42 (50.6%) patients were treated with the 3D-CRT technique, and 41 (49.4%) with the VMAT technique. The patients’ characteristics are summarized in Table 1. All patients treated with 3D-CRT had intermediate-risk prostate cancer, while in the VMAT group the patients had more heterogeneous diseases, including 2 (4.9%) patients with low-risk, 28 (68.3%) with intermediate-risk, and 11 (26.8%) with high-risk cancer. Table 2 outlines treatment characteristics and dose constraints.

The median follow-up time for rectal and genito–urinary toxicity was 77.26 months for the whole group (IQR 67.89; 120.33) (3D-CRT: median time (IQR): 120.39 (107.28;127.27); VMAT: median time (IQR): 68.65 (63.22;72.46)).

Table 3 summarizes the incidence of acute and late treatment related side effects between groups.

Grade 1–2 acute GU toxicity was commonly described among groups: 59.5% of patients in the 3D-CRT group and 68.3% in the VMAT arm. There was no significant difference in acute toxicity rate among groups. Acute GI toxicity grade ≥ 2 was not observed.

The probability of late GI toxicity (grade ≥ 2) has been estimated at 0.0% (0.0 CIF) at 24 months for VMAT and at 2.4% (0.024 CIF) for 3D-CRT (0.32 Gray’s test), at 9.52% (0.0952 CIF) at 60 months for 3D-CRT, and at 4.9% (0.049 CIF) for VMAT (0.41 Gray’s test).

The probability of late genito–urinary toxicity (grade ≥ 2) has been estimated at 0.0% (0.0 CIF) at 24 months for VMAT and at 2.4% (0.024 CIF) for the 3D-CRT method (0.32 Gray’s test), at 4.76% (0.048 CIF) at 60 months for 3D-CRT, and at 0.00% (0.00 CIF) for the VMAT method (0.16 Gray’s test).

As evidenced in Table 4, VMAT allowed, in the large majority of cases, for a dose reduction to the rectum and bladder. Nonetheless, the only parameter that was correlated with clinical toxicity was rectal V66. In fact, GI toxicity events grade ≥ 2 were statistically related with rectal dose limit V66 > 8.5% (*p* = 0.045).

A higher, although not statistically significant, incidence of rectal toxicity was detected for patients undergoing hormone therapy (6.3% vs. 25.4% (0.06 vs. 0.25 CIF); 0.09 Gray’s test at the end of follow-up).

The volume of treatment (PTV) stratified by treatment technique was not related with the risk to develop late or acute toxicity.

The median follow-up time for relapse was 97.78 months (IQR 72.72;116.45) (3D-CRT method: median time (IQR): 116.45 (108.39;128.15); VMAT method: median time (IQR): 72.72 (66.05;76.87)); the median follow-up time for cancer-specific survival was 81.50 months (IQR 72.72;116.45) (3D-CRT method: median time (IQR): 119.64 (109.45;128.15); VMAT method: median time (IQR): 72.72 (67.43;76.44)).

The probability of relapse was estimated at 2.4% (0.02 CIF) at 60 months for 3D-CRT and at 14.6% (0.15 CIF) for the VMAT method (0.045 Gray’s test). At the end of follow-up, we obtained an estimate of 9.6% (0.1 CIF) for 3D-CRT and 14.6% (0.15 CIF) for VMAT considering the whole follow-up period (0.07 Gray’s test).

No statistically significant difference in cancer-specific survival has been reported between the two groups.

The cancer-specific survival has been estimated at 100.0% (CIF 0.00) for 3D-CRT, at 100% (CIF 0.00) for VMAT at 24 months (1 Gray’s test); at 100.0% (CIF 0.00) for 3D-CRT, and at 95.12% (CIF 0.049) for VMAT at 60 months (0.15 Gray’s test).

Diabetes, hypertension, and anticoagulant therapy were not risk factor for GI and GU toxicity or relapse.

## 4. Discussion

Over the last twenty years, randomized clinical trials have demonstrated the benefit of radiation therapy in the treatment of high–intermediate-risk prostate cancer, with the implementation of dose escalation and moderately hypofractionated schedules that allowed for an optimal disease control. Adjuvant hormonal therapy and modern radiotherapy are the cornerstone of this success [12,13]. Hypofractionated dose escalation is an excellent strategy, providing an outstanding increase in the therapeutic index in the treatment of localized prostate cancer [12,13,26]. A total dose of 70 Gy delivered in 28 sessions over 6 weeks, 2.5 Gy per fraction, corresponds to a 80 Gy dose equivalent delivered with conventional fractionation (EQD_2_), accepting the hypothesis that the prostate cancer α/β ratio is 1.5 Gy [3,27]. Alongside this, 70 Gy in 28 fractions corresponds to EQD_2_ 77 Gy for rectum and bladder, assuming a 3–5 Gy α/β ratio.

Treatment-related side effects are defined as toxicities occurring as a direct consequence of therapy. Nevertheless, it is very complicated to assess what the absolute role of radiotherapy in this process is because there are multiple concurring factors associated with adverse events, particularly late toxicity. Possible prognostic factors involved in the assessment of toxicity risk include patient characteristics (such as age at treatment, pre-existing symptoms, and co-morbidities), treatment features (target volume definition, dose per fraction, radiotherapy technique, duration of hormonal therapy), and lastly, the score system criteria utilized for defining acute and late toxicities. Differences in study design and patient selection criteria made it extremely complex to compare clinical trials and retrospective studies on radiation-induced late toxicity. So far, the role of image-guided radiotherapy techniques, moderate hypofractionation, and dose escalation as prognostic factors influencing late GU and GI toxicity is not well defined.

This phase II study aimed to evaluate the impact of hypofractionated radiotherapy delivered with the advanced IGRT technique on treatment-related side effects. Moreover, we compared image-guided radiotherapy 3D-CRT versus IG-VMAT to assess the possible impact of the planning technique on toxicity.

Considering the total cohort, the incidence of severe side effects was extremely low, with acute GU grade ≥ 3 toxicities reported by only 1.2% of patients, and late GU and GI grade ≥ 3 toxicities reported by 1.2% of patients. No acute GI grade ≥ 3 toxicities were reported. Remarkably, grade ≥ 3 toxicities were reported only in the 3D-CRT group, and G2 toxicity rates were as well higher in the 3D-CRT group, although statistical significance could not be reached. Moreover, it must be noted that the VMAT treatments delivered a dose of 70 Gy to a larger volume, including the prostate and the seminal vesicles, while in the 3D-CRT treatments the prescribed dose to the seminal vesicles was of 50 Gy.

As already demonstrated by Kupelian [28], the reduced incidence of side effects in our study could be explained by FM-based IGRT, which allows for online daily repositioning of the target. This procedure makes it possible to minimize the CTV–PTV expansion to 5 mm and, consequently, to reduce of the overlap between PTV and organs at risk.

We also believe that the IGRT procedure itself plays an important role in adverse events reduction: in fact, all the patients included in this analysis presented a good toxicity profile, despite the VMAT technique providing a better dose conformation over 3D-CRT.

The univariate analysis highlighted the correlation among irradiated rectum volume and late adverse events: there was a statistically significant increase in toxicity when 8.5% of rectal volume received 66 Gy or more (V66 > 8.5%). Additionally, most of the other dosimetric parameters were improved by the adoption of VMAT. Nevertheless, the univariate analysis did not show a statistically significant divergence in the risk of toxicity among the analyzed treatment modalities. The correlation between IGRT–VMAT and a lower, although not significant, risk of toxicity compared with IGRT–3DCRT could be explained by the greater conformation of the dose around the PTV volume and the lower dose to the surrounding organs.

Simultaneously, the shorter time taken to perform treatment with the VMAT technique reduces the intra-fraction organ motion and consequently improves the accuracy of the treatment delivery, reducing the chance of accidentally over-dosing the organs at risk.

A recent analysis of the pattern of practice across Italian radiation oncology centers in the 2004–2011 period [29] reported an increased use of IGRT and conformal techniques, confirming the perceived benefits by the clinicians.

No conclusions could be made regarding the prognostic impact of different planning techniques, as 3D-CRT cohorts substantially differed in terms of risk group and ADT use: the V-MAT group included patients with significantly worse NCCN risk group (26.8% high-risk patients vs. 0% in 3D-CRT group) and a higher fraction (26.8% vs. 11.9%) of patients not undergoing ADT. This could explain the slight, although significantly higher, risk of biochemical relapse at 5 years in the VMAT group.

Nonetheless, the increase in risk was not significant considering the whole follow-up, and no significant differences in CSS have been observed between the two groups.

The limits of this study are the small sample size and the heterogeneous characteristics, remarkably regarding dose prescription, tumor risk group, and ADT use of the patients in the VMAT and 3D-CRT group.

Moreover, the management and classification of urologic cancers is relentlessly evolving and multiple innovations and emerging tools that were not yet adopted in our cohort have been introduced in clinical practice. The adoption of the 2014 International Society of Urological Pathology (ISUP) modified Gleason grades [30] is currently the standard of care for the histopathologic assessment of prostate cancer and, in the future, additional molecular markers could aid the clinician in the diagnosis and risk classification [31].

Prostate cancer is a heterogeneous disease, and an accurate molecular stratification could optimize patient selection in order to define a specific therapeutic path tailored to the biological characteristics of neoplastic cells. This approach could be used to design clinical trials [32], and in some instances commercially available tools (e.g., validated assays for the androgen receptor splice variant 7, AR-V7) are already used to guide therapeutic decisions [33].

In recent years, multiple tools have been developed to classify patients at higher risk of relapse and monitor disease control and eventual progression, including liquid biopsy and novel immunohistochemical assays.

Promising results were reported for the adoption of circulating biomarkers for urologic cancers, which could for instance allow for the optimization of the prognostic stratification [34] and the selection of bladder cancer patients that could benefit from adjuvant treatment intensification [35] or to assess the response of renal cell carcinoma to checkpoint inhibitors [36,37].

The most established forms of liquid biopsy are circulating tumor cells (CTCs) and circulating tumor DNA (ctDNA), with a large body of evidence demonstrating their potential predictive and prognostic value in prostate cancer [38,39]. Liquid biopsy may allow for a non-invasive stratification of the tumor characteristics, providing the possibility to delineate a dynamic profile of genomic, epigenomic, and proteomic features evolving over time [39]. Blood-based liquid biopsies have been mostly developed in metastatic settings due to the higher yield of CTCs and ctDNA, but also exhibited promising results in localized tumors as a mean to offer diagnostic signatures aiding in risk stratification and disease monitoring. Other molecules, such as cytokines and other inflammatory mediators, could also be used as circulating biomarkers to predict clinical outcomes and treatment resistance [40].

The main limit for the widespread adoption of liquid biopsy and other circulating biomarkers in the routine clinical practice is the absence of validation and definition of protocols and methodological standards [38,39].

The increasing integration of these innovative tools in the therapeutic workflow should be considered for the design of new clinical trials and studies.

## 5. Conclusions

This study shows that moderate hypofractionation is feasible and safe in patients with intermediate- and high-risk prostate cancer. Daily image-guided radiotherapy may be crucial in decreasing acute and late toxicity by limiting the total volume of treatment and the overlap between PTV and OAR. Moreover, the reduced risk of PTV geographical missing offered by IGRT could improve the disease control rates. The reliability of IGRT is confirmed by the high correlation rate among different operators, which can be further improved by the application of fiducial markers. The adoption of VMAT allows for promising results in terms of OAR sparing and a reduction in toxicity that, also given the small sample, did not reach statistical significance.

## Figures and Tables

**Table 1 jcm-11-06913-t001:** Patients’ characteristics: IG-3DCRT = image-guided 3D conformal radiation therapy; IG-VMAT = image-guided volumetric modulated arc therapy; PSA = prostate-specific antigen.

	3D	VM	Overall	*p*-Value Fisher/Chisq
	(*N* = 42)	(*N* = 41)	(*N* = 83)	
**Age**				
<70 yrs	13.0 (31.0%)	9.00 (22.0%)	22.0 (26.5%)	0.4964
≥70 yrs	29.0 (69.0%)	32.0 (78.0%)	61.0 (73.5%)	
**Diabetes Mellitus**				
No	38.0 (90.5%)	32.0 (78.0%)	70.0 (84.3%)	0.1415
Yes	4.00 (9.5%)	9.00 (22.0%)	13.0 (15.7%)	
**Anticoagulant therapy**				
No	24.0 (57.1%)	20.0 (48.8%)	44.0 (53.0%)	0.5125
Yes	18.0 (42.9%)	21.0 (51.2%)	39.0 (47.0%)	
**Hypertension**				
No	15.0 (35.7%)	8.00 (19.5%)	23.0 (27.7%)	0.1412
Yes	27.0 (64.3%)	33.0 (80.5%)	60.0 (72.3%)	
**NCCN risk group**				**<0.001**
Low	0 (0%)	2.00 (4.9%)	2.00 (2.4%)	
Intermediate	42.0 (100%)	28.0 (68.3%)	70.0 (84.3%)	
High	0 (0%)	11.0 (26.8%)	11.0 (13.3%)	
**Androgen deprivation therapy**				
No	5.00 (11.9%)	11.0 (26.8%)	16.0 (19.3%)	0.1015
Yes	37.0 (88.1%)	30.0 (73.2%)	67.0 (80.7%)	
**Pre-treatment PSA**				0.07996
<10 ng/mL	30.0 (71.4%)	32.0 (78.0%)	62.0 (74.7%)	
10–20 ng/mL	12.0 (28.6%)	6.00 (14.6%)	18.0 (21.7%)	
>20 ng/mL	0 (0%)	3.00 (7.3%)	3.00 (3.6%)	
**Clinical T stage**				0.7779
T1c-T2a	23.0 (56.1%)	26.0 (61.9%)	49.0 (59.0%)	
T2b	10.0 (24.4%)	11.0 (26.2%)	21.0 (25.3%)	
>T2b	7.00 (17.1%)	5.00 (11.9%)	12.0 (14.5%)	
Missing	1.00 (2.4%)	0 (0%)	1.00 (1.2%)	
**Gleason score**				**0.01757**
≤6	11.0 (26.8%)	13.0 (31.0%)	24.0 (28.9%)	
7	23.0 (56.1%)	29.0 (69.0%)	52.0 (62.7%)	
8–10	7.00 (17.1%)	0 (0%)	7.00 (8.4%)	

**Table 2 jcm-11-06913-t002:** Treatment characteristics and dose constraints.

	IG-3DCRT	IG-VMAT
**PTV total dose**	PTV1= 50Gy	PTV = 70Gy
	PTV2= 20Gy	
**PTV prescription isodose**	V95% = 100%	V100% ≥ 95%
		D2% < 108%
		D98% > 94%
**Bladder**	V50 ≤ 30% D_max_ < 70 Gy	V52 < 30%
		V62 < 10%
		D1% < 70 Gy
**Rectum**	V60 ≤ 30% D_max_ < 70 Gy	V46 < 30%
		V52 < 17%
		D1% < 70 Gy
**Femur Head**	V50 < 5%	V30 < 5%
		Daverage < 20 Gy
**Penile Bulb**	D90% ≤ 50 Gy	D_average_ < 20 Gy
	D70% ≤ 70 Gy	

**Table 3 jcm-11-06913-t003:** Incidence of acute and late adverse events.

	3D-CRT	VMAT	Overall	*p*-Value Fisher
	(*N* = 42)	(*N* = 41)	(*N* = 83)	
**Acute GU toxicity**				0.3111
G0	16.0 (38.1%)	13.0 (31.7%)	29.0 (34.9%)	
G1	20.0 (47.6%)	26.0 (63.4%)	46.0 (55.4%)	
G2	5.00 (11.9%)	2.00 (4.9%)	7.00 (8.4%)	
G3	1.00 (2.4%)	0 (0%)	1.00 (1.2%)	
**Acute GI toxicity**				1
G0	37.0 (88.1%)	36.0 (87.8%)	73.0 (88.0%)	
G1	5.00 (11.9%)	5.00 (12.2%)	10.0 (12.0%)	
**Late GU toxicity**				1
G0	36.0 (85.7%)	37.0 (90.2%)	73.0 (88.0%)	
G1	4.00 (9.5%)	4.00 (9.8%)	8.00 (9.6%)	
G2	1.00 (2.4%)	0 (0%)	1.00 (1.2%)	
G3	1.00 (2.4%)	0 (0%)	1.00 (1.2%)	
**Late GI toxicity**				0.7883
G0	32.0 (76.2%)	33.0 (80.5%)	65.0 (78.3%)	
G1	6.00 (14.3%)	6.00 (14.6%)	12.0 (14.5%)	
G2	4.00 (9.5%)	2.00 (4.9%)	6.00 (7.2%)	

**Table 4 jcm-11-06913-t004:** Outlines of rectal and bladder dosimetric volume histogram (DVH) for radiotherapy technique.

	3D-CRT	VMAT	Overall	*p*-Value Wilcoxon Mann–Whitney Test/*t* Test
	(*N* = 42)	(*N* = 41)	(*N* = 83)	
**Rectal V40**				0.4327
Mean (SD)	35.7 (8.67)	37.1 (7.37)	36.4 (8.04)	
Median [Min, Max]	35.7 [18.2, 58.8]	36.6 [23.3, 60.0]	36.2 [18.2, 60.0]	
**Rectal V45**				**0.027**
Mean (SD)	27.4 (7.33)	24.5 (3.51)	25.9 (5.91)	
Median [Min, Max]	27.6 [13.3, 44.4]	24.5 [17.0, 31.1]	25.4 [13.3, 44.4]	
**Rectal V66**				**0.0012**
Mean (SD)	6.32 (3.33)	4.04 (1.75)	5.19 (2.89)	
Median [Min, Max]	6.03 [0.0920, 14.9]	4.20 [0.0300, 7.80]	4.90 [0.0300, 14.9]	
**Rectal mean dose**				0.5327
Mean (SD)	33.3 (4.63)	33.8 (7.59)	33.5 (6.24)	
Median [Min, Max]	34.4 [22.6, 42.8]	33.0 [24.4, 74.1]	33.3 [22.6, 74.1]	
**Bladder V40**				**<0.001**
Mean (SD)	30.2 (12.6)	18.3 (11.1)	24.3 (13.2)	
Median [Min, Max]	30.2 [7.79, 65.9]	15.2 [5.20, 44.3]	23.8 [5.20, 65.9]	
**Bladder V66**				**0.0016**
Mean (SD)	7.50 (4.53)	4.81 (3.20)	6.17 (4.14)	
Median [Min, Max]	6.79 [1.13, 21.0]	3.80 [1.30, 15.6]	5.05 [1.13, 21.0]	
**Bladder mean dose**				**0.0014**
Mean (SD)	27.8 (8.19)	20.2 (12.1)	24.1 (10.9)	
Median [Min, Max]	27.9 [9.74, 48.5]	17.2 [8.20, 74.6]	24.0 [8.20, 74.6]	

## Data Availability

The data that support the findings of this study are available from the corresponding author, upon reasonable request.

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
