# Peer review of "Prostate Cancer Treatment-Related Toxicity: Comparison between 3D-Conformal Radiation Therapy (3D-CRT) and Volumetric Modulated Arc Therapy (VMAT) Techniques"

_jcm, 2022, doi:10.3390/jcm11236913_

Round 1

Reviewer 1 Report

The authors describe outcomes for two modalities of RT for prostate cancer. Overall the work is scientifically sound and the message is clear cut. Methods and conclusions are reasonable. 

Major comments.

In the method section the authors state that their survival analysis include two cox regressions - these are not correspondingly addressed in the results section. The Kaplan Meier estimates are however. Since (like the authors acknowledge) the cohorts differ - an unadjusted model like the KM method is probably not that important to include. One suggestion would be to remove the survival analysis (both the cox and the KM test) from the manuscript.

Minor comments:

1. Line 79. t looks like you excluded patients with high functionality should < really be >?

2. Remove the quotation marks on the hospital name.

3. Tables.  Table 3 is inadequate. The abbreviation PSA is explained although not in the table at all whereas the concentration of something (testosterone, PSA?) is. The substrata of "No" is included under the heading "Yes" which is confusing. Furthermore, the categorical exposure-variables diabetes, hypertension etcetera should be presented with yes or no instead of 0/1 which is more suitable for the worksheet in your statistical software.

Author Response

We would like to thank the Reviewer for the insightful analysis and for the valuable comments.

Major comments.

In the method section the authors state that their survival analysis include two cox regressions - these are not correspondingly addressed in the results section. The Kaplan Meier estimates are however. Since (like the authors acknowledge) the cohorts differ - an unadjusted model like the KM method is probably not that important to include. One suggestion would be to remove the survival analysis (both the cox and the KM test) from the manuscript.

As suggested, we removed data regarding survival from the text.

Minor comments:

1. Line 79. It looks like you excluded patients with high functionality should < really be >?

We corrected the typing error

2. Remove the quotation marks on the hospital name.

Quotation marks have been removed

3. Tables. Table 3 is inadequate. The abbreviation PSA is explained although not in the table at all whereas the concentration of something (testosterone, PSA?) is. The substrata of "No" is included under the heading "Yes" which is confusing. Furthermore, the categorical exposure-variables diabetes, hypertension etcetera should be presented with yes or no instead of 0/1 which is more suitable for the worksheet in your statistical software.

The wrong format of the table has been included in the proof. We apologize for the mistake, that has been fixed.

Reviewer 2 Report

In this manuscript the authors aimed to to compare gastrointestinal (GI) and genitourinary (GU) toxicity rates among PCa pts who underwent radical RT using 3D Conformal Radiation Therapy (3D-CRT) or VMAT approach. They found that moderate hypofractionation is feasible and safe in pts with intermediate and high-risk PCa. Moreover, no statistically significant differences in overall and cancer-specific survival have been reported between the two groups. 

Overall interesting results in a still under-investigated topic of uro-oncological field.

Following some comments:

1) Any information about variant histologies PCa? In 2016 Humphrey et al. described a novel classification of urological malignancies in which variant histologies have been found be correlated with aggressive behaviour and advanced clinical stage. See for example doi: 10.1038/s41391-022-00600-y.

2) Any information oabout prostate volume and/or prostate tumor volume?

3) Since the authors further investigated the survival outcomes (within a large timeframe) they should consider to discuss in a separate paragraph - and as a point of future perspective - the role of emerging tools for disease monitoring. The stable adoption of - for example - liquid biopsy, novel immunohistochemical assays, seems to be promising across prostate, renal, and urothelial carcinoma as well the response to novel ARTAs, PARP-i or immune check piont inhibitors, or conventional chemotherpy agents. Nowdays a multisciplinary view is mandatory among uro-oncological manuscripts. The manuscript would benefit from such parallelism with prostate cancer itself (Actas Urol Esp (Engl Ed). 2020 Apr;44(3):139-147. doi: 10.1016/j.acuro.2019.08.007), (Eur Urol. 2021 Jun;79(6):762-771. doi: 10.1016/j.eururo.2020.12.037), (Prostate. 2022 Nov;82(15):1456-1461. doi: 10.1002/pros.24419. Epub 2022 Jul 28.), (N Engl J Med. 2015 Oct 29;373(18):1697-708) (Eur Urol. 2018 Apr;73(4):572-582. doi: 10.1016/j.eururo.2017.10.036. Epub 2017 Nov 10.), bladder cancer (Eur Urol Oncol. 2021 May 6;S2588-9311(21)00078-X. doi: 10.1016/j.euo.2021.04.004), (Asian J Urol. 2021 Oct;8(4):376-390. doi:10.1016/j.ajur.2021.05.001),  (Eur Urol Oncol. 2021 Apr;4(2):204-214. doi: 10.1016/j.euo.2020.01.003),  (Urol Oncol. 2022 Mar;40(3):110.e1-110.e9. doi: 10.1016/j.urolonc.2021.10.010. Epub 2021 Dec 11. ), and renal cell carcinoma (Curr Drug Targets. 2020;21(16):1664-1671. doi: 10.2174/1389450121666200324151056.), (Lancet Oncol. 2022 May;23(5):612-624. doi: 10.1016/S1470-2045(22)00128-0. Epub 2022 Apr 4.)

Author Response

Overall interesting results in a still under-investigated topic of uro-oncological field.

We would like to thank the Reviewer for the interest and for his precious suggestions, that allowed us to improve the quality of the manuscript.

Following some comments:

1) Any information about variant histologies PCa? In 2016 Humphrey et al. described a novel classification of urological malignancies in which variant histologies have been found be correlated with aggressive behaviour and advanced clinical stage. See for example doi: 10.1038/s41391-022-00600-y.

Patients have been enrolled between 2008 and 2014, when this classification was not implemented yet. We added the role of novel classifications in the ‘future perspective’ section.

2) Any information about prostate volume and/or prostate tumor volume?

Unfortunately, this information was not reported.

3) Since the authors further investigated the survival outcomes (within a large timeframe) they should consider to discuss in a separate paragraph - and as a point of future perspective - the role of emerging tools for disease monitoring. The stable adoption of - for example - liquid biopsy, novel immunohistochemical assays, seems to be promising across prostate, renal, and urothelial carcinoma as well the response to novel ARTAs, PARP-i or immune check piont inhibitors, or conventional chemotherpy agents. Nowdays a multisciplinary view is mandatory among uro-oncological manuscripts. The manuscript would benefit from such parallelism with prostate cancer itself.

We would like to thank the Reviewer for this valuable suggestion. A new paragraph regarding future perspective, including the valuable suggested references, has been added to the manuscript.

Round 2

Reviewer 2 Report

The authors revised the manuscript fully and properly. Congratulations.